# Phytosterol–γ-Oryzanol–Glycerol Monostearate Composite Gelators for Palm Stearin/Linseed Oil Oleogel-Based Margarine: Nutrient Enrichment, Textural Modulation, and Commercial Product Mimicry

**DOI:** 10.3390/foods14071206

**Published:** 2025-03-29

**Authors:** Jingwen Li, Yujuan Hu, Qing Ma, Dongkun Zhao, Xinjing Dou, Baocheng Xu, Lili Liu

**Affiliations:** 1College of Food and Bioengineering, Henan University of Science and Technology, Luoyang 471003, China; jingwen2268928436@163.com (J.L.); hyj1231232023@163.com (Y.H.); m15137666063@163.com (Q.M.); kun18238796961@163.com (D.Z.); douxj521@163.com (X.D.); yangliuyilang@126.com (L.L.); 2Henan International Joint Laboratory of Food Green Processing and Safety Control, Luoyang 471000, China; 3National Experimental Teaching Demonstration Center of Food Processing and Safety, Henan University of Science and Technology, Luoyang 471003, China

**Keywords:** margarine, oleogel, linseed oil, phytosterol, solid fat substitution

## Abstract

This study prepared palm stearin/linseed oil-based margarines (PST/LO-BMs) and palm stearin/linseed oil oleogel-based margarines (PST/LO-OBM) by incorporating varying proportions (20–60% oil phase) of linseed oil (LO) and LO-based oleogel, respectively. By comparing PST/LO-OBMs and PST/LO-BMs, it was found that the introduction of phytosterol–γ-oryzanol (PO) complexes and glycerol monostearate (GMS) to PST/LO-OBM induced three distinct crystalline morphologies: needle-like crystals, spherical crystals, and cluster-type crystals. These crystal assemblies synergistically constructed a robust three-dimensional network, effectively entrapping both aqueous droplets and liquid oil fractions while substantially reinforcing the structural integrity of PST/LO-OBM. Notably, the incorporated gelators modified the crystallization behavior, where GMS likely served as a nucleating site promoting triglyceride crystallization. This structural modulation yielded favorable β’-form crystal polymorphism, which is critically associated with enhanced textural properties. Comparative analysis with commercial margarine revealed that the PST45/LO40-OBM formulation exhibited comparable rheological performance, crystalline type, and thermal properties, while demonstrating superior nutritional characteristics, containing elevated levels of α-linolenic acid (23.54%), phytosterol (1410 mg/100 g), and γ-oryzanol (2110 mg/100 g). These findings provide fundamental insights for margarine alternatives with nutritional attributes.

## 1. Introduction

Margarine represents a water-in-oil emulsion, typically comprising 80–85% oil phase, 17–14% aqueous phase, and minor emulsifier components [1]. The oil phase incorporates a critical proportion of solid fats that establish a crystalline network structure, effectively immobilizing liquid oils and finely dispersed water droplets [2]. In current commercial production, these solid fats predominantly originate from two sources: hydrogenated vegetable oils containing trans fatty acids (TFAs) and animal-derived fats (e.g., beef tallow and lard) rich in saturated fatty acids (SFAs) [3]. Extensive research has established conclusive associations between excessive consumption of both SFAs and TFAs with increased risks of cardiovascular pathologies, cerebrovascular accidents, and coronary heart disease [4]. This substantiates the imperative for developing nutritionally optimized solid fat alternatives to replace conventional margarine formulations. The ideal substitutes should demonstrate significantly reduced SFA and TFA content while providing enhanced concentrations of unsaturated fatty acids (UFAs) and beneficial nutraceutical components.

Oleogels have emerged as innovative solid fat substitutes [5], with peer-reviewed studies confirming their dual capacity to replicate the functional properties of traditional solid fats while eliminating trans fatty acids (TFAs) and typically containing elevated unsaturated fatty acid (UFA) content [6]. The structural foundation of these systems relies on the formation of a thermoreversible crystalline network, where structuring agents immobilize liquid oils through three-dimensional molecular entrapment, achieving the desired semi-solid rheology [7]. The phytosterol–γ-oryzanol (PO) binary system has garnered significant research interest as a co-structuring agent, particularly due to its unique self-assembling network formation and demonstrated nutraceutical benefits, including cardioprotective effects and potential chemopreventive properties [8]. Glycerol monostearate (GMS) has also gained widespread application in oleogel formulations owing to its unique ability to impart distinctive textural and structural properties to reduce solid fat content (SFC) oil products [9]. Emerging applications demonstrate successful PO- and GMS-based oleogel integration in lipid-based food matrices, with documented efficacy in cocoa butter analogs [10], chocolate spread [11], and other commercial products requiring solid fat replacement strategies.

Nevertheless, a notable research gap persists regarding the application of PO-oleogels and GMS-oleogels as solid fat substitutes in margarine formulations, compared to their applications in other foods. Current research efforts have predominantly concentrated on developing natural wax-derived oleogel systems for margarine production. Notable examples include the utilization of sunflower wax, rice bran wax, beeswax, and candelilla wax to create wax–hempseed oleogel matrices as solid fat replacements in margarine systems [12]. While these wax-based formulations demonstrate promising textural properties—achieving comparable hardness to commercial spreadable margarine at <3% gelling agent concentration—a critical limitation remains the persistent wax-derived aftertaste negatively impacting sensory perception. To address this organoleptic challenge, advanced formulation strategies have emerged. An innovative approach combines beeswax with Chinese lacquer wax in equimolar ratios to create eutectic mixtures [13], effectively modulating the crystallization behavior to produce wax-based margarines with improved mouthfeel characteristics and visual/textural parity to commercial counterparts [1]. The technological viability of these modified systems has been further validated through functional food applications. Comparative baking trials revealed that cookies formulated with optimized beeswax-based margarine (Biscuit A) maintained equivalent textural parameters and sensory attributes to those prepared with conventional margarine (Biscuit B), while demonstrating superior nutritional profiles through significant reductions in both trans fatty acids (TFAs) and saturated fatty acids (SFAs) [14].

Notably, existing research has confirmed that both the type of gelator and the triglyceride molecular architecture within the oil phase critically determine the physicochemical characteristics of the resultant oleogels [8]. Building on this foundation, we developed a novel oleogel-based margarine formulation distinct from wax oleogel-based margarine systems. Specifically, PO and GMS were employed as composite gelators, while linseed oil (LO) and palm stearin (PST) constituted the oil phase to develop palm stearin/linseed oil oleogel-based margarines (PST/LO-OBMs). The PST/LO-OBM formulation integrates three nutritional considerations: (i) LO serves as an optimal source of α-linolenic acid, providing well-documented cardiovascular benefits [15]; (ii) PST modulates plasticity through its characteristic solid fat content (SFC) profile while optimizing the fatty acid balance of the margarine [16]; (iii) PS exhibits lowering cardiovascular disease risk and potential anticancer properties [17,18,19]. However, in this complex system, it remains unclear whether the gelling agent and the oil phase can interact synergistically to form a stable crystalline network that can evenly distribute the water droplets and liquid oil droplets, whether an excessive network density might impede margarine melting, and whether multi-component crystallization could enhance β’-form development to meet the desired texture properties. To address these, we comparatively analyzed palm stearin/linseed oil-based margarine (PST/LO-BM) and oleogel-structured margarine (PST/LO-OBM), focusing on melting behavior, polymorphism, microstructure, and texture properties. Additionally, by optimizing the PST/LO oleogel ratio, we developed a nutritionally enhanced oleogel-structured margarine with texture, rheological, thermal properties, and appearance comparable to those of commercial margarine. These findings provide the rational design of functional margarine products.

## 2. Materials and Methods

### 2.1. Materials and Chemicals

First-grade pressed linseed oil was purchased from the Inner Mongolia Yishanyuan Biotechnology Co., Ltd. (Ulanqab, China). Palm stearin was purchased from Yihai Kerry Food Technology Co., Ltd. (Tianjin, China). Phytosterol (95%) was purchased from Xi’an Tianbao Biotechnology Co., Ltd. (Xi’an, China). γ-oryzanol (98%) and glycerol monostearate were purchased from Shanghai Yuanye Biotechnology Co., Ltd. (Shanghai, China). Skimmed milk powder was purchased from Fonterra Trading (Shanghai) Co., Ltd. (Shanghai, China). Soy lecithin was purchased from Shanghai Aladdin Biochemical Technology Co., Ltd. (Shanghai, China).

### 2.2. Preparation of Oleogel

A binary structuring system comprising phytosterol–γ-oryzanol (PO) and glycerol monostearate (GMS) at an 8:2 (*w/w*) ratio was homogenized with linseed oil through thermomechanical processing. The mixture underwent controlled heating (80 °C ± 1 °C) with continuous magnetic stirring (650 rpm, DF-101S, Gongyi Yuhua instrument Co., Ltd., Gongyi, China) until the complete dissolution of the structurants. Subsequent gelation of the mixture was achieved through controlled crystallization at 4 °C for 24 h in sealed containers to obtain an oleogel.

### 2.3. Preparation of Margarine

The preparation of margarines (PST/LO-OBM and PST/LO-BM) was developed following Chai et al. [1] with critical modifications. (i) For the aqueous phase (14.7%), 0.2 wt% salt and 2.0 wt% skimmed milk powder were dissolved in deionized water under vortex mixing (650 rpm, 8 min). (ii) For the oil phase (85.3%), precisely formulated blends containing 0.3 wt% soy lecithin, palm stearin (PST), and oleogel substitutions (20–60% *w/w*) were homogenized at 650 rpm (65 °C). (iii) For emulsification, phase combination (aqueous phase and oil phase) followed by high-shear processing (8000 rpm, 3 min, 65 °C) were employed. (iv) For crystallization, rapid chilling in an ice-water bath with gentle agitation was carried out, followed by structural maturation (24 h, 25 °C) and final storage at 4 °C before analysis. The composition of the PST/LO-BM and PST/LO-OBM samples and their name code are shown in Table 1.

### 2.4. Texture Properties

The firmness and adhesiveness of the margarine samples were determined using a TA.XT texture analyzer (SMS TA.XT Express Enhanced, Stable Micro Systems Ltd., Surrey, UK) with a P/2.0 probe. The test parameters were a pre-test velocity of 5.0 mm/s, a mid-test velocity of 1.0 mm/s, a post-test velocity of 5.0 mm/s, a trigger force of 2.0 g, and a total compression distance of 10.0 mm. Each sample was placed in a beaker (25 mL volume, diameter 30 mm × 55 mm height), ensuring that the sample interface was flat and that the distance between the probe and the sample surface was kept consistent. Each sample was measured in triplicate.

### 2.5. Rheological Properties

The rheological properties of the margarine samples were determined using a TA DHR-2 rheometer (TA DHR-2, Waters Corporation, Milford, MA, USA) equipped with a 40 mm diameter parallel plate, and a gap of 1000 μm was employed during the measurement. A cold-water circulation system (the work temperature was set to 15 °C) was used to control the temperature.

The rheological characterization of all margarine samples was conducted at 25 °C. Initially, amplitude sweep tests were performed at a constant frequency of 1 Hz with strain amplitudes ranging from 0.01% to 100% to determine the viscoelastic properties through storage modulus (G’) and loss modulus (G”) measurements. Subsequently, frequency sweep analyses were executed under controlled deformation conditions: strain amplitude was maintained at 0.1% (within the predetermined linear viscoelastic region, LVR) while applying oscillatory frequencies from 0.1 to 10 Hz, enabling the comprehensive characterization of the frequency-dependent viscoelastic behavior through dynamic mechanical analysis.

### 2.6. Appearance and Microstructure

The microstructures of the margarine samples were observed using optical microscopy and a polarized light microscope (PLM, XPV-203, Changfang Optical Instrument Co., Ltd., Shanghai, China). The margarine samples analyzed by optical microscopy (OLM) and a polarized light microscope were prepared as follows. Firstly, a small amount of margarine sample was placed on a glass slide; then, a coverslip was placed on the surface of the sample and gently pressed to evenly distribute the sample into a thin layer. In addition, photographs were taken to record the color and appearance of these samples.

Subsequently, the margarine samples were further observed using a laser confocal microscope (CLSM). These samples were stained with Nile red solution (concentration of 0.1%) and stored in the dark, and the microstructure of the samples was observed at an excitation wavelength of 518 nm. The magnification of OLM, PLM, and CLSM was 100×.

### 2.7. Polymorphism

The polymorphic structure of the margarine samples was determined by X-ray diffraction (XRD) (SmartLab SE, Rigaku Corporation, Tokyo, Japan). The sample was scanned from 5° to 40° at a rate of 2°/min at ambient temperature.

### 2.8. Thermal Analysis

The thermal properties of the margarine samples were analyzed using a DSCQ2000 differential scanning calorimeter (DSCQ2000, New Castle, DE, USA). A total of 10 mg of the margarine sample was placed into an aluminum crucible and sealed. The sealed empty aluminum crucible served as a control. The program was set as follows: firstly, the margarine sample was heated to 80 °C and held for 10 min until it was completely melted to remove its crystalline memory; then, the temperature was rapidly decreased to 0 °C and kept for 10 min to reach equilibrium; finally, the temperature was increased to 80 °C at a rate of 5 °C/min to analyze its melting behavior.

### 2.9. Solid Fat Content

The solid fat content (SFC) of the margarine samples was analyzed using pulsed nuclear magnetic resonance (Minispec mq 20, Bruker, Billerica, MA, USA). Before determination, the standard substance (triolein) was first used to calibrate the temperature. After calibration, approximately 3 g of sample was placed in pNMR tubes and heated in a water bath at 90 °C for 30 min to eliminate crystal memory. Then, the samples were cooled to 0 °C and held for 1 h to allow for their complete crystallization. Subsequently, the SFC of the samples was measured at different temperatures, ranging from 0 °C to 60 °C, with a 5 °C increase at each step.

### 2.10. Fatty Acid Composition

Fatty acid composition and individual fatty acid content were determined using the procedure described in GB 5009.168-2016 [20]. Firstly, 0.1000 g margarine sample and 0.8 mL of 2mol/L potassium hydroxide methanol solution were added into a 10 mL glass test tube to obtain mixture A. Then, mixture A was subjected to ultrasonication at 50 °C for 10 min. Subsequently, 4 mL of n-hexane was added to the test tube to obtain mixture B. We then vortexed mixture B thoroughly and collected the supernatant after standing. Then, 0.5 g of anhydrous sodium sulfate was added into the collected supernatant and vortexed for 30 s to remove moisture. Finally, the n-hexane phase was centrifuged, and we collected the supernatant for subsequent analysis. The fatty acid was determined based on the relative retention time of the fatty acid methyl ester standard, and we used the normalization method to calculate the relative percentage content of each fatty acid.

### 2.11. Statistical Analysis

All experiments and measurements were carried out in triplicate. The statistical analyses of variance were carried out using SPSS 26.0. The statistical differences were obtained by a one-way analysis of variance (ANOVA) using Duncan model testing.

## 3. Results and Discussion

### 3.1. Fatty Acid Composition of Oleogel-Based Margarine

The fatty acid compositions of PST, LO, and their blended oleogel-based margarines (PST/LO-OBMs) are shown in Table 2. PST is dominated by saturated fatty acids (SFAs), with palmitic acid (C16:0) accounting for 50.664% of its total fatty acids. In contrast, LO contains 85.121% unsaturated fatty acids (UFAs), primarily oleic (C18:1), linoleic (C18:2), and linolenic acids (C18:3). By adjusting the LO-based oleogel content from 60% to 20%, the SFA content in PST/LO-OBMs increased from 28.871% to 58.079%, while UFA decreased proportionally, from 71.129% to 41.921%. This demonstrated that the fatty acid composition of PST/LO-OBMs could be tailored through adjustment of the LO/PST ratio.

The crystallization behavior and thermal properties of lipid systems are critically influenced by fatty acid composition and triglyceride (TAG) structure [21]. PST, rich in long-chain SFAs (C16:0), predominantly comprises high-melting-point TAGs such as 12.53% of SSS (melting point: 54–65 °C) and 48.70% SUS (melting point: 27–42 °C) [16,22], which promote dense crystalline networks and elevate crystallization temperatures [23]. Conversely, LO-based oleogel contains TAGs with unsaturated bonds (e.g., 70.90% of UUU (melting point: −14–1 °C) and 20.64% of SUU (melting point: 1–23 °C)) [22,24], whose kinked structures hinder molecular packing, resulting in lower melting points and delayed crystallization [25]. Consequently, reducing the LO oleogel content in PST/LO-OBM decreases the proportion of low-melting-point unsaturated TAGs while increasing high-melting-point saturated TAGs. This shifted the PST/LO-OBM’s melting profile toward higher temperatures, as evidenced by the thermal analysis of PST/LO-OBMs. These findings highlight the tunability of PST/LO-OBM properties through the blending of SFAs and UFAs, balancing thermal stability for margarine applications.

Furthermore, the fatty acid profile of commercial margarine is analyzed, revealing the following composition: C14:0 (1.046%), C16:0 (46.920%), C18:0 (6.814%), C18:1 (32.858%), C18:2 (11.456%), C18:3 (0.573%), and C20:0 (0.334%). SFAs and UFAs constituted 55.114% and 44.886% of the total fatty acids, respectively. In contrast, PST/LO-OBMs demonstrated a significantly elevated UFA content (particularly linolenic acid, C18:3) compared to commercial margarine. If we were to take PST45/LO40-OBM as an example, its UFA content and linolenic acid were 16.325% and 22.958% higher than those of commercial margarine, respectively. In addition to fatty acid modulation, PST/LO-OBMs contain bioactive compounds such as PS and γ-oryzanol, which contribute to additional health benefits, including lipid-lowering effects, cardiovascular protection, and anticancer properties [8,17,19].

### 3.2. Texture Properties of Oleogel-Based Margarine

Firmness is an important index to evaluate the quality of margarine, which determines its culinary performance and spreadability [26]. As demonstrated in Figure 1A, a marked increase in the firmness of PST/LO-BM and PST/LO-OBM was observed, as the LO and LO-based oleogel content decreased from 60% to 20%. Specifically, PST/LO-BM showed an increase in firmness from 4.59 g to 250.11 g, while that of PST/LO-OBM rose from 55.65 g to 450.32 g. This enhanced firmness might be attributed to the increase in the PST content. The higher PST content could provide a greater solid fat content and promote the formation of more triglyceride crystals, thereby enhancing the firmness of the margarine. By comparison, the increase in firmness of PST/LO-OBMs was more pronounced than that recorded for PST/LO-BMs. In addition, by comparing the firmness of PST/LO-BMs and PST/LO-OBMs with that of commercial margarine, it was found that PST65/LO20-BM containing 65% PST exhibited a markedly lower firmness than commercial margarine, while PST55/LO30-OBM containing 55% PST demonstrated comparable firmness. These differences might be attributed to the distinct crystallization behaviors of triglycerides and gelators [13,27]. In contrast to PST/LO-BMs, containing only small triglyceride crystals, the PST/LO-OBM contained spherical crystals, needle crystals, and cluster crystals. It is the co-existence of multiple crystal types that leads to the formation of a dense network structure in PST/LO-OBMs, which successfully binds liquid oil and water droplets, thus significantly increasing the firmness of PST/LO-OBM. Similarly, in the study of beeswax-based margarine, the firmness of margarine was attributed to the gelators (beeswax) and increased with a higher content of beeswax [1]. Kato Rondou et al. [28] also demonstrated that incorporating gelators, such as berry wax, candelilla wax, and carnauba wax, could enhance liposome system firmness by modulating PST crystallization behavior.

Adhesiveness, a key indicator of margarine mouthfeel, quantitatively reflects the stickiness of food products through the negative force area measured during the first compression cycle [26]. As demonstrated in Figure 1B, both PST/LO-BM and PST/LO-OBM formulations exhibited a remarkable increase in adhesiveness as the LO and LO-based oleogel content decreased from 60% to 20%. Specifically, PST/LO-BM adhesiveness increased from 3.27 g to 95.52 g, while PST/LO-OBM showed an even more pronounced increase from 28.98 g to 145.58 g. This inverse correlation between LO or LO-based oleogel content and adhesiveness was probably because of the reduced lubrication effect, that is, the lower liquid oil content decreased the interfacial slippage between margarine surfaces and the probe, thereby enhancing the perceived stickiness. The observed trend was consistent with their firmness, suggesting that the formation of denser crystalline networks in low LO or LO-based oleogel margarine formulations showed stronger adhesiveness. Consistent with our finding, Jia Hu et al. [29] demonstrated a positive correlation between adhesiveness (417.20–521.73 g) and hardness (139.61–206.16 g) in four bigel-based margarine formulations (A–D). The comparatively higher adhesiveness in their formulations (approximately 4–5 times greater than our maximum values) could be attributed to the distinctive viscoelastic properties imparted by the potato starch–water hydrogel component, whose hydrophilic polymers enhanced interfacial adhesion through water-mediated hydrogen bonding. Notably, all PST/LO-BMs and PST/LO-OBMs exhibited significantly lower adhesiveness than commercial margarine (163.92 g), implying that PST/LO-BMs and PST/LO-OBMs spend less energy to overcome the stickiness of margarine for chewing compared to commercial margarine [30].

### 3.3. Rheological Property of Oleogel-Based Margarine

As shown in Figure 2, the rheological properties of two types of laboratory-made margarines (PST/LO-BM and PST/LO-OBM) were compared with those of commercial margarine (CM). The results of the static shear scan in Figure 2A–C show that the apparent viscosity of all samples decreased with an increasing shear rate, exhibiting shear-thinning behavior. This was because, as the shear rate increased, the molecular forces within the internal structure of the margarine depolymerized, and the molecules aligned directionally along the flow, resulting in reduced apparent viscosity [31].

Viscoelastic characterization is a critical index of rheological testing. The storage modulus (G′) represents elastic (solid-like) behavior, while the loss modulus (G″) quantifies the viscous (liquid-like) response [32]. As shown in Figure 2D–F, the dynamic oscillatory sweep results of the PST/LO-BM, PST/LO-OBM, and CM samples all showed a predominant elastic character (G’ > G’‘) within the linear viscoelastic region (LVR). Both G’ and G’‘ decreased with increasing strain, indicating progressive structural weakening of the margarines under larger deformations, possibly leading to network disintegration [33]. Furthermore, the viscoelastic properties of PST/LO-BM and PST/LO-OBM were significantly influenced by their fat structure [34]. Specifically, as the LO and LO-based oleogel content decreased from 60% to 20%, accompanied by a gradual increase in the PST content with a high solid fat content (SFC), both G’ and G’‘ in PST/LO-BM and PST/LO-OBM increased. Notably, compared to PST/LO-BM, the addition of gelators was associated with a remarkable rise in both G’ and G’‘ and the yield stress of PST/LO-OBM.

The dynamic frequency sweep results (Figure 2G–I) revealed that PST/LO-BM, PST/LO-OBM, and CM all did not show frequency dependence within the tested range, and G’ was significantly higher than G’‘, indicating that all the samples formed network structure and behaved as if they had a solid state. With the decrease in LO and LO-based oleogel content from 60% to 20%, their network strength increased, while the yield stress values decreased in both PST/LO-BM and PST/LO-OBM samples. Notably, the network strength of PST/LO-OBM increased more significantly than that of PST/LO-BM, indicating that the addition of gelators greatly enhanced the network strength of the margarine. This observation was consistent with their measured firmness. In addition, to achieve a network strength comparable to commercial margarine, 55% PST was needed for PST/LO-BM, while the proportion of PST in PST/LO-OBM required could be reduced from 55% to 35% by adding gelators. It was observed that, when the PST content was added at 35% and 45% (corresponding to 50% and 40% LO-based oleogel content, respectively), the resulting PST/LO-OBM exhibited a network strength similar to that of commercial margarine. Notably, these two margarines exhibited significantly higher yield stress values compared to those of commercial margarine, suggesting enhanced plasticity.

### 3.4. Appearance and Microstructure of Oleogel-Based Margarine

Figure 3 presents the comparative optical light microscopy (OLM) results of PST/LO-BMs, PST/LO-OBMs, and commercial margarine. Most PST/LO-BMs exhibited large water droplets (indicated by red circles) and coalesced lipid phases. By contrast, PST/LO-OBMs demonstrated smaller water droplet and oil droplet dimensions after the addition of gelators. This contrast was particularly evident in the PST35/LO50-BM and PST35/LO50-OBM systems, attributable to their distinct crystalline structures. As shown in Figure 4a, the PST35/LO50-BM system formed a crystalline network entrapping liquid oil aggregates and water droplets (red circles). This crystalline network originated from the coalescence of smaller spherical crystalline units [28]. Conversely, the PST35/LO50-OBM system in Figure 4b exhibited a hybrid network composed of water droplets (indicated by red circles) and liquid oil droplets embedded within a crystalline matrix. This structural configuration was corroborated by CLSM analysis, which demonstrated superior droplet uniformity. The network of PST35/LO50-OBM in Figure 4b comprises three characteristic crystals: (i) small spherical PST crystals (green rectangle), (ii) needle-like crystals (blue rectangle), and (iii) clustered crystal aggregates (yellow rectangle) forming a continuous three-dimensional network. This morphological diversity arises from PO/GMS-mediated modulation of crystallization processes: (i) PS disrupted GMS lamellar crystal formation through molecular interference [35] and promoted the formation of small spherical GMS crystals. These spherical crystals likely acted as nucleation sites for PST crystals and accelerated the formation of PST triglyceride crystals [36]. (ii) Uncomplexed PS formed characteristic needle crystals due to its intrinsic crystallization propensity [35]. (iii) PS–γ-oryzanol hydrogen bonding facilitated liquid oil encapsulation, subsequently inducing cluster crystal assembly via intermolecular interactions [37]. The composite crystalline structure of PST35/LO50-OBM provided enhanced mechanical performance, exhibiting a 96.3 g increase in firmness compared to PST35/LO50-BM, along with an order of magnitude increase in the viscoelastic moduli (G′/G″). These improvements stemmed from the gelator-induced densification of crystalline networks, which optimized phase distribution and structural integrity.

Notably, PST/LO-OBMs maintained stability through an interconnected three-dimensional network that effectively entrapped and uniformly dispersed both aqueous phases and yellow-colored liquid oils. This structural feature imparted a yellow hue to the PST/LO-OBM sample, contrasting with the paler coloration of commercial margarine. The enhanced yellow coloration observed in PST/LO-OBMs originated from intrinsic phytopigments naturally present in the vegetable oil, whereas the paler coloration of commercial margarine was attributable to its higher dairy fat content, in contrast to the exclusively plant-derived oil in PST/LO-OBMs. Additionally, the structural configuration of PST/LO-OBMs also maintained integrity, showing no observable phase separation of aqueous or lipid components (Figure 3C). Microstructural analysis (PLM and CLSM in Figure 4 and Figure A1) revealed that PST/LO-OBM demonstrated a microstructure comprising multiple crystalline structures, whereas commercial margarine exhibited a homogeneous crystalline structure with limited structural diversity. Despite these differences, both PST/LO-OBMs and commercial margarine achieved effective droplet stabilization.

### 3.5. Polymorphism of Oleogel-Based Margarine

In an oil system, there are three typical types of polymorphic forms: α, β, and β’, which are generated based on the different growth environments of the crystals [21]. The polymorphic forms of PST/LO-OBM and PST/LO-BM are shown in Figure 5. Four types of diffraction peaks can be observed in both margarine samples, with short spacings of 4.6, 4.35, 4.2, and 3.8, respectively. Among them, the peak with a short spacing of 4.6 corresponds to the β crystalline form, while the peaks with short spacings of 4.35, 4.2 and 3.8 correspond to β’ crystalline forms. According to the theory, β and β’ crystal forms co-exist in PST/LO-BM and PST/LO-OBM. This is because PST has heterogeneous triglycerides, which tend to undergo different polymorphic event [36].

However, by comparing the polymorphism results of PST/LO-BM and PST/LO-OBM, it was observed that the diffraction peaks corresponding to the β’ crystalline forms were significantly enhanced by the addition of the gelators. This finding suggested that the incorporation of the gelators could promote the formation of small-sized β’ triglyceride crystals of PST. This might be attributed to the higher melting temperature of GMS compared to PST. During rapid cooling, GMS has a tendency to form nuclei earlier, which could accelerate the overall crystallization process of the GMS-PST blend [36]. Therefore, the early nucleation of GMS likely provided a structural template that facilitated the co-crystallization of PST, resulting in the enhanced formation of β’ crystalline forms. This result aligns with previous investigations focusing on gelling agent-mediated modifications of PST crystallization dynamics. Notably, compared to a pure PST system, the carnauba wax–PST system revealed distinct nucleation behavior: (i) carnauba wax addition induced PST crystallite refinement and (ii) carnauba wax crystals were formed first, subsequently functioning as nucleation templates for PST crystal growth through epitaxial growth processes in the carnauba wax–PST blends [28]. As we all know, it has been reported that the β’ crystal is an ideal crystal form, as it has a small size, a large specific surface area, and can easily form a dense crystal network. In addition, the β’ crystal has moderate firmness and melts easily at oral temperatures [38]. Thus, PST/LO-OBM could provide a smooth mouthfeel, like commercial margarine, because both had β and β’ crystalline forms, with β’ crystalline forms being more predominant than the β crystalline form (Figure 5B,C).

### 3.6. Melting Property of Oleogel-Based Margarine

As we all know, solid fat content (SFC) values at different temperatures are usually measured to understand the melting properties of fats and oils. The SFC curves of PST/LO-BM and PST/LO-OBM are shown in Figure 6D–F. By comparing the SFC of PST/LO-BM and PST/LO-OBM (Figure 6D,E), it was observed that their SFC decreased as the temperature increased. However, PST/LO-OBM always had a higher SFC than PST/LO-BM when the proportion of LO in PST/LO-BM and LO-based oleogel in PST/LO-OBM increased from 20% to 60%. This was likely because the addition of oleogel agents could lead to a tighter crystals network (including triglyceride crystals and crystals formed by gelators) in margarine [1]. This tight network slowed the melting of the oil crystals, resulting in a higher solid fat content (SFC) at the same temperature [13]. Notably, when other conditions were same, the melting point of PST/LO-OBMs was nearly the same as that of PST/LO-BMs (both samples reached SFC = 0 when temperature ≥ 55 °C), Figure 6D,E), although the addition of the gelling agent caused a slight increase in the SFC of PST/LO-OBMs compared to PST/LO-BMs.

In order to further investigate the melting characteristics of margarine (PST/LO-BMs, PST/LO-OBMs, and commercial margarine), we determined their melting curves from 0 to 80 °C using differential scanning calorimetry (DSC), as shown in Figure 6D–F. The peak onset temperature, peak temperature, and enthalpy change in the samples are summarized in Table 3. Comparative thermal analysis revealed three distinct endothermic peaks (Peaks 1, 2, and 3) in both the PST/LO-BM and PST/LO-OBM systems (Figure 6A,B), but their peak temperatures and enthalpies were different. Specifically, there was a small melting peak (Peak 1) with a broad shoulder for PST/LO-BMs (peak temperatures of 7.79–13.08 °C) and PST/LO-OBMs (peak temperatures of 9.44—12.89 °C), which represented the crystallization process of the components consisting of SUU-type TAGs [22]. The comparison of peak temperatures of Peak 1 of PST/LO-BMs and PST/LO-OBMs revealed that the addition of the gelling agent did not significantly change the peak temperature ranges of this peak 1. Similarly, the peak temperatures of Peak 3 in PST/LO-BMs and PST/LO-OBMs also did not change remarkably (PST/LO-BMs: 48.06–53.11 °C vs. PST/LO-OBMs: 50.85–54.55 °C), but the enthalpy in PST/LO-OBMs was lower than in PST/LO-BMs. However, Peak 2 in PST/LO-BMs changed from a flat melting peak (enthalpy from 0.93 to 10.77 J/g) to a sharp melting peak (enthalpy from 16.60 to 26.71 J/g) after the addition of the gelling agent. The melting peak temperature of Peak 2 increased to a higher temperature (33.83–45.58 °C in PST/LO-BMs vs. 43.91–48.24 °C in PST/LO-OBMs). This change in thermodynamic behavior suggests that the addition of the gelators (PO and GMS) altered the crystallization of high-melting TAGs (Peak 2 and Peak 3) of the margarine, expressing as later-onset crystal network disintegration during the heating cycles. A similar phenomenon was found in a previous study [36], whereby the addition of GMS easily accelerated the crystallization of high-melting-point TAG (higher than 21 °C) in PST better than that of low-melting-point TAG in PST due to stronger intermolecular forces [23]. Importantly, the addition of these gelling agents altered the crystallization behavior of PST/LO-OBMs, resulting in a melting profile more similar to that of commercial margarine (Peak 1: 10.51 °C, 17.33 J/g; Peak 2: 43.35 °C, 21.84 J/g; Peak 3: 50.34 °C, 3.91 J/g) after the addition of these gelators (Figure 6B,C).

## 4. Conclusions

In this study, PST/LO-BMs and PST/LO-OBMs were prepared with varying ratios of LO and LO-based oleogel (20%, 30%, 40%, 50%, and 60% of oil phase), respectively. Distinct microstructural differences emerged between PST/LO-BM and PST/LO-OBM: while PST/LO-BM exclusively contained a spherical triglyceride crystalline network formed through PST crystallization, PST/LO-OBM exhibited three distinct crystalline components—(i) needle-like PS crystals, (ii) spherical structures likely resulting from GMS-PST co-crystallization, and (iii) clustered PO-derived crystals formed via self-assembly mechanisms. This polymorphic crystallization behavior suggests that the incorporated gelators induced a denser crystal network within the PST/LO-OBM. The dense architecture of PST/LO-OBM demonstrated dual functionality: effective entrapment of aqueous droplets and liquid lipid, coupled with significant mechanical reinforcement, as evidenced by enhanced firmness. Furthermore, the gelling additives adjusted the triglyceride crystallization behavior and the polymorphic distribution type. This phenomenon might be attributed to the nucleating effect of high-melting-point GMS, which promoted the crystallization of GMS-PST complexes and facilitated the formation of β’ type crystals. These crystal polymorphs were associated with improved organoleptic properties, manifested through reduced droplet size and enhanced phase homogeneity. Notably, the PST/LO-OBM formulation containing a 40% LO-based oleogel content displayed similar rheological properties, network strength, crystal distribution, and thermal property to commercial margarine. Crucially, this optimized formulation demonstrated superior plasticity at ambient temperature (25 °C) compared to commercial margarine, while delivering enhanced nutritional value through elevated α-linolenic acid (23.535%), phytosterol (1410 mg/100 g), and γ-oryzanol (2110 mg/100 g) concentrations. These findings provide fundamental insights for margarine alternatives with nutritional attributes.

## Figures and Tables

**Figure 1 foods-14-01206-f001:**
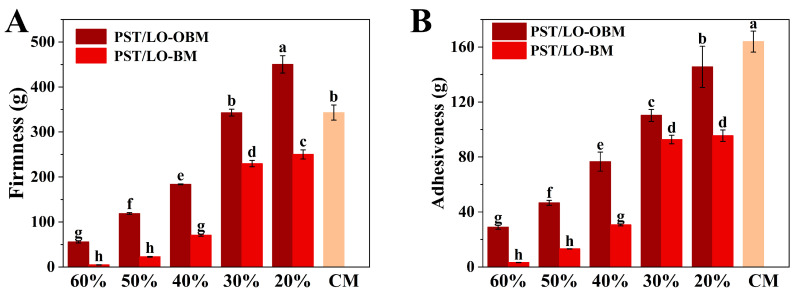
(**A**) Firmness of PST/LO-OBM, PST/LO-BM, and commercial margarine (CM). (**B**) Adhesiveness of PST/LO-OBM, PST/LO-BM, and commercial margarine Different lowercase letters indicate significant differences (*n* = 3, *p* < 0.05). The 60%, 50%, 40%, 30%, and 20% represent the proportions of LO and LO-based oleogel added in PST/LO-BM and PST/LO-OBM, respectively.

**Figure 2 foods-14-01206-f002:**
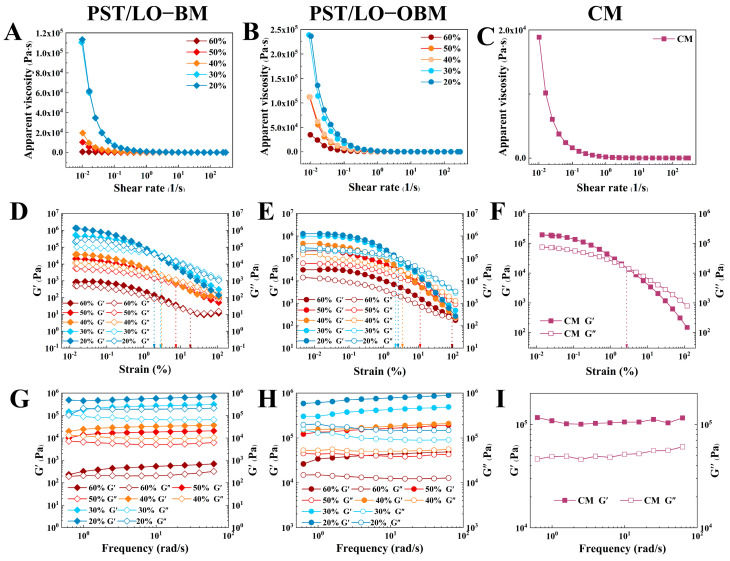
Rheological property of PST/LO-BM, PST/LO-OBM, and commercial margarine (CM): (**A**–**C**) static shear sweep; (**D**–**F**) dynamic oscillatory sweep; and (**G**–**I**) dynamic frequency sweep. The 60%, 50%, 40%, 30%, and 20% represent the proportions of LO and LO-based oleogel added in PST/LO-BM and PST/LO-OBM, respectively.

**Figure 3 foods-14-01206-f003:**
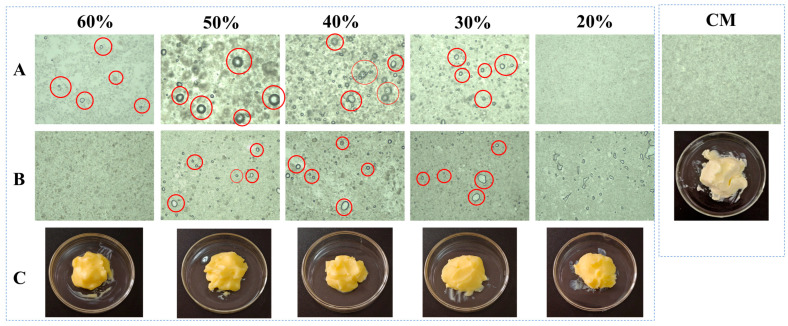
Appearance and optical microscopy microstructure of PST/LO-BM, PST/LO-OBM, and commercial margarine (CM). (**A**) optical microscopy microstructure images of PST/LO-BMs; (**B**) optical microscopy microstructure images of PST/LO-OBMs; (**C**) appearance of PST/LO-OBMs. The 60%, 50%, 40%, 30%, and 20% represent the proportions of LO and LO-based oleogel added to PST/LO-BM and PST/LO-OBM, respectively. The red circles highlight water droplets observed in the sample.

**Figure 4 foods-14-01206-f004:**
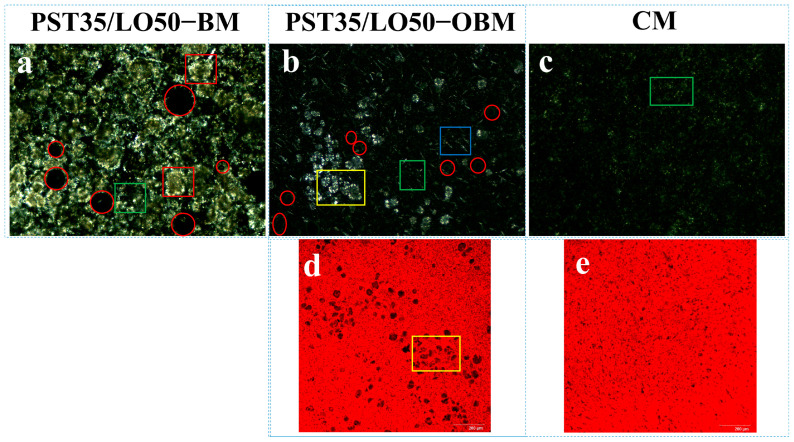
Polarized light microscopy and laser confocal microscopy microstructure of PST35/LO50-BM, PST35/LO50-OBM, and commercial margarine (CM): (**a**–**c**) polarized light microscopy microstructure results of PST35/LO50-BM, PST35/LO50-OBM, and commercial margarine (CM), respectively; (**d**,**e**) laser confocal microscopy results of PST35/LO50-OBM and commercial margarine (CM), respectively. The legend is defined as follows: water droplets (highlighted in red circles); small spherical PST crystals (green rectangles); large spherical PST crystal aggregates (red rectangles); needle-like PS crystals (blue rectangles); and clustered crystals (yellow rectangles).

**Figure 5 foods-14-01206-f005:**
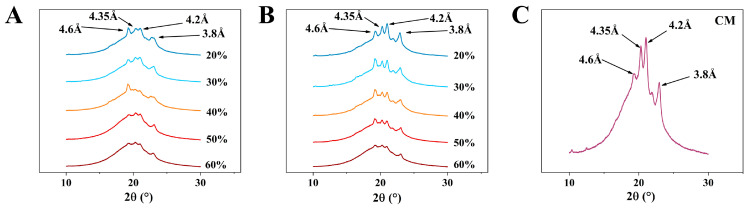
Polymorphism of PST/LO-BM, PST/LO-OBM, and commercial margarine (CM): (**A**–**C**) polymorphism results of PST/LO-BM, PST/LO-OBM, and CM, respectively. The 60%, 50%, 40%, 30%, and 20% represent LO and LO-based oleogel added in proportions of 60%, 50%, 40%, 30%, and 20% to PST/LO-BM and PST/LO-OBM, respectively.

**Figure 6 foods-14-01206-f006:**
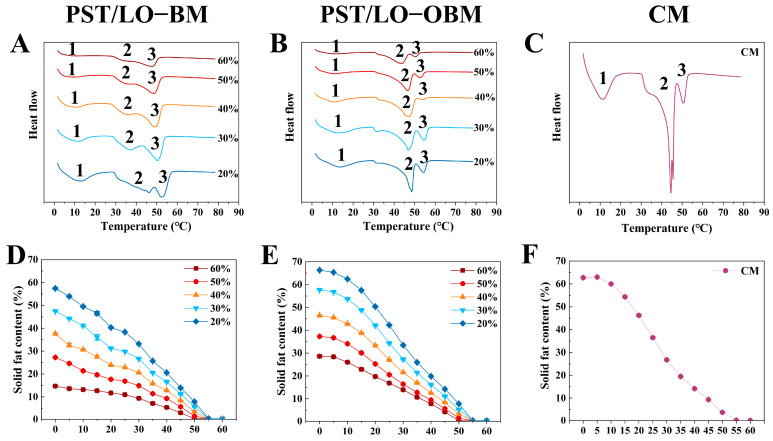
Melting property (**A**–**C**) and SFC (**D**–**F**) of PST/LO-BM, PST/LO-OBM, and commercial margarine (CM). The 60%, 50%, 40%, 30%, and 20% represent that LO and LO-based oleogel were added in proportions of 60%, 50%, 40%, 30%, and 20% to PST/LO-BM and PST/LO-OBM, respectively.

**Table 1 foods-14-01206-t001:** The formulations of PST/LO-BMs and PST/LO-OBMs.

Margarine	Sample	Oil Phase (wt%)						Aqueous Phase (wt%)		
		PST	LO	Soy Lecithin	GMS	PS	γ-Oryzanol	Water	Skimmed Milk Powder	Salt
PST/LO-BM	PST65/LO20-BM	65	20	0.3	-	-	-	12.5	2.0	0.2
	PST55/LO30-BM	55	30	0.3	-	-	-	12.5	2.0	0.2
	PST45/LO40-BM	45	40	0.3	-	-	-	12.5	2.0	0.2
	PST35/LO50-BM	35	50	0.3	-	-	-	12.5	2.0	0.2
	PST25/LO60-BM	25	60	0.3	-	-	-	12.5	2.0	0.2
PST/LO-OBM	PST65/LO20-OBM	65	17.8	0.3	0.44	0.70	1.06	12.5	2.0	0.2
	PST55/LO30-OBM	55	26.7	0.3	0.66	1.06	1.58	12.5	2.0	0.2
	PST45/LO40-OBM	45	35.6	0.3	0.88	1.41	2.11	12.5	2.0	0.2
	PST35/LO50-OBM	35	44.5	0.3	1.10	1.76	2.64	12.5	2.0	0.2
	PST25/LO60-OBM	25	53.4	0.3	1.32	2.11	3.17	12.5	2.0	0.2

PST/LO-BM: palm stearin/linseed oil-based margarine; PST/LO-OBM: palm stearin/linseed oil oleogel-based margarine; PST: palm stearin; LO: linseed oil; GMS: glycerol monostearate; and PS: phytosterol.

**Table 2 foods-14-01206-t002:** Fatty acid composition of PST, LO, and PST/LO-OBMs.

FA (%)	PST	LO	PST25/LO60 -OBM	PST35/LO50 -OBM	PST45/LO40 -OBM	PST55/LO30 -OBM	PST65/LO20 -OBM
C14:0	1.295 ± 0.003 ^a^	0.133 ± 0.005 ^e^	0.491 ± 0.113 ^d^	0.495 ± 0.067 ^d^	0.543 ± 0.015 ^d^	0.800 ± 0.005 ^c^	0.928 ± 0.023 ^b^
C16:0	50.664 ± 0.724 ^a^	7.664 ± 0.228 ^g^	21.017 ± 0.586 ^f^	26.928 ± 0.202 ^e^	32.961 ± 0.924 ^d^	42.894 ± 0.454 ^c^	50.208 ± 0.226 ^b^
C18:0	7.986 ± 0.239 ^a^	6.854 ± 0.641 ^b^	6.904 ± 0.422 ^b^	7.661 ± 0.520 ^ab^	7.044 ± 0.243 ^ab^	7.340 ± 0.007 ^ab^	6.722 ± 0.021 ^b^
C18:1	31.151 ± 1.145 ^a^	22.591 ± 1.455 ^c^	22.709 ± 1.394 ^c^	23.524 ± 0.619 ^c^	26.077 ± 0.912 ^b^	26.413± 0.184 ^b^	29.218 ± 0.141 ^a^
C18:2	7.938 ± 0.167 ^d^	19.183 ± 1.249 ^a^	15.734 ± 0.988 ^b^	12.646 ± 0.181 ^c^	11.604 ± 0.388 ^c^	8.792 ±0.106 ^d^	7.806 ± 0.064 ^d^
C18:3	0.475 ± 0.011 ^g^	43.347 ± 3.547 ^a^	32.687 ± 0.551 ^b^	28.549 ± 0.001 ^c^	23.531 ± 0.17 ^d^	13.513 ± 0.160 ^e^	4.897 ± 0.052 ^f^
C20:0	0.492 ± 0.023 ^a^	0.227 ± 0.021 ^b^	0.459 ± 0.164 ^a^	0.199 ± 0.012 ^b^	0.343 ± 0.026 ^ab^	0.248 ± 0.004 ^b^	0.221 ± 0.028 ^b^
SFA	60.437 ± 0.989 ^a^	15.184 ± 0.411 ^g^	28.871 ± 0.957 ^f^	35.505 ± 1.117 ^e^	40.891 ± 1.156 ^d^	51.282 ± 0.450 ^c^	58.079 ± 0.257 ^b^
USFA	39.563 ± 0.989 ^f^	85.121 ± 0.843 ^a^	71.129 ± 0.957 ^b^	64.495 ± 1.117 ^c^	61.211 ± 1.816 ^d^	48.718 ± 0.450 ^e^	41.921 ± 0.257 ^f^

All values are the mean ± SD of three analyses. FA, fatty acid; PST, palm stearin; LO, linseed oil; SFA: saturated fatty acid; and USFA: unsaturated fatty acid. Different lowercase letters in the same line indicate significant differences (*n* = 3, *p* < 0.05).

**Table 3 foods-14-01206-t003:** Thermodynamic parameters of PST/LO-BMs and PST/LO-OBMs.

Samples		PST25/LO60-BM	PST35/LO50-BM	PST45/LO40-BM	PST55/LO30-BM	PST65/LO20-BM
Peak 1	T_o_/°C	1.74 ± 0.04 ^bc^	1.68 ± 0.04 ^c^	1.72 ± 0.01 ^c^	1.82 ± 0.02 ^ab^	1.86 ± 0.04 ^a^
	T_p_/°C	7.79 ± 0.35 ^b^	7.92 ± 0.71 ^b^	8.13 ± 1.00 ^b^	11.35 ± 1.10 ^a^	13.08 ± 0.39 ^a^
	ΔH, J/g	2.95 ± 0.43 ^d^	6.33 ± 0.29 ^c^	8.65 ± 0.07 ^c^	11.66 ± 2.23 ^b^	19.78 ± 1.31 ^a^
Peak 2	T_o_/°C	28.38 ± 0.02 ^a^	28.33 ± 0.37 ^a^	28.35 ± 0.20 ^a^	28.76 ± 0.22 ^a^	29.02 ± 0.39 ^a^
	T_p_/°C	33.83 ± 0.49 ^c^	34.16 ± 0.02 ^c^	34.49 ± 0.38 ^c^	37.95 ± 1.09 ^b^	45.58 ± 1.50 ^a^
	ΔH, J/g	0.93 ± 0.07 ^d^	2.42 ± 0.21 ^cd^	4.38 ± 0.60 ^c^	8.21 ± 0.92 ^b^	10.77 ± 1.61 ^a^
Peak 3	T_o_/°C	39.11 ± 0.30 ^d^	39.63 ± 0.47 ^d^	41.14 ± 0.06 ^c^	43.30 ± 0.54 ^b^	49.25 ± 0.59 ^a^
	T_p_/°C	48.06 ± 0.16 ^d^	48.68 ± 0.14 ^d^	49.72 ± 0.10 ^c^	50.71 ± 0.02 ^b^	53.11 ± 0.67 ^a^
	ΔH J/g	5.49 ± 0.05 ^d^	9.53 ± 0.35 ^b^	13.69 ± 0.81 ^a^	13.32 ± 0.08 ^a^	7.66 ± 0.55 ^c^
Samples		PST25/LO60-OBM	PST35/LO50-OBM	PST45/LO40-OBM	PST55/LO30-OBM	PST65/LO20-OBM
Peak 1	T_o_/°C	1.81 ± 0.13 ^a^	1.84 ± 0.18 ^a^	1.81 ± 0.13 ^a^	1.80 ± 0.04 ^a^	1.79 ± 0.03 ^a^
	T_p_/°C	9.44 ± 0.59 ^c^	10.86 ± 0.70 ^bc^	10.87 ± 0.58 ^bc^	12.06 ± 0.52 ^ab^	12.89 ± 1.17 ^a^
	ΔH, J/g	8.60 ± 0.39 ^d^	14.84 ± 0.23 ^c^	16.17 ± 0.21 ^c^	19.66 ± 1.92 ^b^	26.83 ± 2.39 ^a^
Peak 2	T_o_/°C	33.20 ± 4.47 ^a^	30.52 ± 0.67 ^a^	32.75 ± 0.23 ^a^	33.29 ± 2.38 ^a^	36.08 ± 0.70 ^a^
	T_p_/°C	43.91 ± 0.42 ^a^	46.55 ± 0.12 ^b^	47.08 ± 0.24 ^ab^	46.81 ± 0.76 ^b^	48.24 ± 0.60 ^a^
	ΔH, J/g	16.60 ± 0.21 ^c^	22.98 ± 0.58 ^b^	24.68 ± 0.74 ^ab^	24.74 ± 0.83 ^ab^	26.71 ± 1.47 ^a^
Peak 3	T_o_/°C	48.18 ± 0.12 ^d^	50.04 ± 0.14 ^c^	51.50 ± 0.02 ^a^	50.11 ± 0.18 ^c^	50.84 ± 0.01 ^b^
	T_p_/°C	50.85 ± 0.08 ^d^	52.62 ± 0.01 ^c^	54.05 ± 0.08 ^b^	54.12 ± 0.01 ^b^	54.55 ± 0.07 ^a^
	ΔH J/g	1.33± 0.03 ^bc^	2.43 ± 0.15 ^b^	0.68 ± 0.33 ^c^	8.18 ± 0.57 ^a^	7.22 ± 1.29 ^a^

T_o_ represents the initial melting temperature of margarines; T_p_ represents the peak melting temperature of margarines; and ΔH represents the enthalpy change in margarines. Different lowercase letters in the same line indicate significant differences (*n* = 3, *p* < 0.05).

## Data Availability

The original contributions presented in this study are included in the article; further inquiries can be directed to the corresponding author.

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
