# Peer review of "Phytosterol–γ-Oryzanol–Glycerol Monostearate Composite Gelators for Palm Stearin/Linseed Oil Oleogel-Based Margarine: Nutrient Enrichment, Textural Modulation, and Commercial Product Mimicry"

_foods, 2025, doi:10.3390/foods14071206_

Round 1

Reviewer 1 Report

Comments and Suggestions for Authors

General comments

The manuscript titled 'Effect of Phytosterol-γ-Oryzanol/Glycerol Monostearate on the Physical and Chemical Properties of Palm Stearin/Linseed Oil Oleogel-based Margarine' was developed from a large set of analyses and presents interesting results. However, the presentation and discussion of the results need to be greatly improved.

1 Acronyms

The text makes excessive use of acronyms. Some of the acronyms are not previously identified in the text when the expression related to them was written for the first time, nor in the list of abbreviations. Regarding acronyms, the authors should:

  • use acronyms only for the identification of their formulations or in situations where the expression associated with the acronym is frequently repeated;
  • ensure that the acronyms used have all the associated words present in the text and in the list of abbreviations;
  • do not use acronyms in the abstract."

2 The figures are too small, which makes their analysis difficult, especially the microscopy images and the curves related to rheological behavior, whose resolution is already poor. The captions should be carefully detailed and presented along with the figure title, carefully specifying which image in the figure it refers to.

  1. The text includes a large number of interesting and useful analyses to explain the studied phenomena. However, the authors use a relatively small number of literature references to corroborate their results and substantiate their explanations. Therefore, a greater number of literature references, and from reputable journals, should be explored for the discussions of the results.

  1. The text is somewhat confusing in some points. It should be rewritten and undergo a review of the English language

Specific comments

  1. Topic 2.3 - Preparation of oleogel-based margarine

The authors must to describe this topic in more detail and carefully. Based on this, they should present in a table the systems (margarines) formulated from the described method. The mass percentages (or fractions) of each major component (linseed oil, palm stearin, oryzanol, phytosterol, and glycerol monostearate) should be placed in the table for each system. Finally, it is suggested that the acronym referring to each formulation incorporate the mass percentages (fractions) of the components that have their quantity altered for each system.

  1. Topic 2.5 Rheological Properties

In this topic, the authors first state: 'The work temperature was set as 15 °C.' Immediately following, there is the statement 'The rheological characterization of all margarine samples was conducted at 25 °C.'

What is the correct temperature? Why was it chosen?

  1. Topic 2.6 Appearence and microstructure

What was the image magnification used in the microscopy analyses?

  1. Topic 3.1. Texture property of oleogel-based margarine

What is the point in reporting the firmness unit in grams? Why didn't the authors report it as consistency? Adhesiveness results were presented, but they were not discussed at any point. The adhesiveness and firmness results should be separated into two different images of the same figure for the discussion of the results.

  1. Topic 3.2 Rheological property of oleogel-based margarine

5.1 The images relating to Figure 2 should be significantly enlarged.

5.2 Replace the term 'viscosity' with 'apparent viscosity' since it is not a Newtonian fluid.

5.3. The term 'ductility' is not applicable to the type of system studied and should be corrected.

5.4. The results of the oscillatory tests are discussed in only 3 lines (258–261). There is much more information that can be extracted from these tests based on the scientific literature. The authors should explore these results much more extensively.

  1. Topic 3.3. Appearance and microstructure of oleogel-based margarine

6.1. The images are too small and of low resolution. The magnification selected for microscopy should be higher. These characteristics do not allow for detailed analysis of the figure. For example, it is not possible to determine the types of cluster and crystal lattice formed. The authors should provide images with higher magnification and/or improve the size and resolution of the images presented.

6.2. With the aid of image processing software, it is possible to obtain, for example, the average size of the crystalline clusters, which is of interest in this type of work.

6.3 The discussion of microscopy must be associated with the results of rheology and consistency (firmness).

6.4. In line 319, it is written "... exhibited enhanced structural complexity through crystal polymorphism...". However, what is being discussed here is microstructure. The evaluation of polymorphism can only be done with the aid of other techniques, such as X-ray diffraction (Topic 3.4).

  1. Topic 3.5. Melting property of oleogel-based margarine

Calorimetry analyses (DSC) provide much more information than just a melting peak temperature. They provide, for example, the energy consumed in the melting of the sample, or the starting and ending temperatures of the melting process. The authors should separate the numerical results of the DSC analysis from the images and present them in a separate table. The other numerical data, in addition to the peak temperature, will allow for a greater discussion on this topic.

  1. Topic 3.6. Fatty Acid Composition of oleogel-based margarine

The fatty acid composition results should be the first topic of the results and discussion, as they affect the entire crystallization process of the systems. These results should be presented in a table.

From literature data, the authors should also obtain the approximate triglyceride composition of the oils present in the system. Based on this composition, a discussion should be made correlating the triglyceride composition (which truly determines the crystallization process) with the behavior of the systems.

Comments on the Quality of English Language

The English language used is understandable; however, some expressions are used incorrectly, especially those more technical ones associated with the content addressed. There are punctuation errors. A review of the English language will make the text clearer and more objective.

Reviewer 2 Report

Comments and Suggestions for Authors

The manuscript Effect of Phytosterol-γ-Oryzanol/Glycerol Monostearate on the Physical and Chemical Properties of Palm Stearin/Linseed Oil Oleogel-based Margarine contains information on the physical and chemical properties of two margarine-type products. It is important to realize that the concentration, type of oil, and characteristics of each minor component definitely influence the functional properties of the final product, in this case, margarine. On the other hand, the selection of a gelator and its interaction with oils are essential to determine the structure, mechanical, and functional properties of oleogels, making them suitable for different applications. In this work, oleogel preparation with linseed oil and a binary structuring system consisting of phytosterol/oryzanol and glycerol monostearate was used to prepare margarine. In general, the manuscript does not clearly establish the innovation of the work; the authors could reorganize the main ideas and state what the novelty and objective of the study are. In addition, the authors should pay attention to minor recommendations to improve the manuscript.

1.-Title has an error typographical, change “Plam stearin” to “Palm stearin”

2.-Introduction: It is mandatory that the relevance and the objective of the work are clearly stated.

3.-Methodology: 1) it is necessary to include a table that contains the different formulations of the two margarines (LO/PST-BM and LO/PST-OBM). This will allow a clear understanding of the discussion of the results (as an example review page 6, lines 267-272). 2) in section 2.6, appearance and microstructure did not include the methodology of the optical microscopy.

  1. Results and discussion. 1) In Figure 1, is missing the texture results of the commercial margarine, 2) On page 9 line 349 what means CRW?, this abbreviation was not defined in the abbreviations section, 3) Why was the phytosterols and oryzanol content only determined in LO/PST/BM with 40% LO based oleogel?, and how was it determined? Are the concentrations of phytosterols (352 mg/100g ) and oryzanol (528 mg/ 100g) in the margarine obtained enough to provide health benefits? the authors should justificate this point. Finally, in the manuscript, the effect of phytosterol-γ-oryzanol/glycerol monostearate on the physical and chemical properties of palm stearin/linseed oil oleogel-based margarine was missing; the authors must pay attention to this point because it the title of the work. The discussion of the results was made in general terms about the effect on the oleogel content in margarine; also, in some sections, it was highlighted that palm stearin had the greatest effect. Then, the effect of phytosterol-γ-oryzanol/glycerol monostearate on human health was not determined in this study.
  2. Figures. Is it possible to improve the quality or size of the figures? The details are missing in the current version

Round 2

Reviewer 2 Report

Comments and Suggestions for Authors

The authors have made all recommendations to enhance the final version of the manuscript.